# The Future of Functional Clothing for an Improved Skin and Textile Microbiome Relationship

**DOI:** 10.3390/microorganisms9061192

**Published:** 2021-05-31

**Authors:** Rosie Broadhead, Laure Craeye, Chris Callewaert

**Affiliations:** Center for Microbial Ecology and Technology, Ghent University, Coupure Links 653, 9000 Ghent, Belgium; Rosanna.Broadhead@UGent.be (R.B.); Laure.Craeye@UGent.be (L.C.)

**Keywords:** skin microbiome, textile microbiome, anti-odour, functional clothing

## Abstract

The skin microbiome has become a hot field of research in the last few years. The emergence of next-generation sequencing has given unprecedented insights into the impact and involvement of microbiota in skin conditions. More and more cosmetics contain probiotics or bacteria as an active ingredient, with or without scientific data. This research is also acknowledged by the textile industry. There has been a more holistic approach on how the skin and textile microbiome interacts and how they influence the pH, moisture content and odour generation. To date, most of the ingredients have a broad-spectrum antibacterial action. This manuscript covers the current research and industry developments in the field of skin and textiles. It explores the nature of antimicrobial finishing in textiles which can disrupt the skin microbiome, and the benefits of more natural and microbiome friendly therapies to combat skin conditions, malodour and skin infection.

## 1. Introduction: Background and Driving Forces

The clothing and fashion industry is a multi-billion market with a fast pace of innovation. In the years 1990 to 2000, laundering habits have shifted from chemical-based detergents and hot washes towards enzyme-based detergents and cold washes, driven by environmental and economic reasons [1]. The side effect of the lower washing temperatures, enzyme-based cleaning technology, decreased water consumption and absence of bleaching agents is a reduced microbial and volatile removal in the laundry process [2]. Polyester and other synthetic fibres have become a mainstream yarn in clothing textiles. On average, two-thirds of new textiles produced are synthetic and over half are made from fossil fuel-based polyester [3]. However, polyester is prone to selective bacterial growth and odour development [2].

Clothing textiles absorb skin sweat and microbes, which can lead to an increased odour generation and bacterial colonisation. Textile microorganisms can cause staining, fabric deterioration and even physical irritation, such as skin allergies or infections [4]. In recent decades, a wide range of textile finishes, antimicrobial techniques, nanoparticle applications and bioactive remedies have been developed to answer this need.

## 2. Skin & Textiles

### 2.1. The Skin

The skin is the largest organ in the human body [5]. Its complex surface and semipermeable membranes interact with the immune system and inner body as well as the outer environment. The ability for water to permeate the skin’s surface can be considered one of the most crucial skin functions that make human life possible [6]. This porous surface is a highly efficient self-repairing barrier designed “to keep in the inside in and outside out” [7]. The skin gives protection against chemical, physical and biological hazards. Chemical hazards include irritating agents, while physical and biological hazards are UV radiation and pathogens or insects, respectively. Yet, the different protective functions of the skin are often linked or even co-regulated [8,9]. In addition, the skin has a regulatory role as well. Examples of this include moisture release, dehydration prevention, blood pressure regulation and body temperature control [8,10]. It is clear that the skin has an indispensable function where its importance can be seen when confronted with skin diseases [8].

Despite the skin’s semi-porous characteristic, it is generally perceived by society as a rigid “symbolic surface between self and world” [11]. This functional and visible interface is regarded by many as the basis of personal identity. As such, skin does not contribute solely to physical properties, but also to emotional human well-being as skin appearance affects one’s self-esteem. The skin’s biology, often regarded as the superficial layer or trivialised as vanity, is now well-recognised as a real interdisciplinary research topic within sciences [6]. In this way, the cultural attitude towards the skin, compared with the physical qualities, are quite contradictory. While society considers skin as an absolute barrier, this major organ is working hard to allow substances to pass through to protect and preserve the body’s equilibrium.

### 2.2. The Skin Microbiome

“Our skin serves as a primary defence, a sensory and an excretory organ” [12]. The skin is also home to an estimated 10^12^ microorganisms that live and feed on the skin secretions and desquamations [12]. These billions of microorganisms, including bacteria, fungi, protists and viruses, form the so-called skin microbiome [13,14]. It is known that these organisms are more numerous in comparison to the body’s own cells: there are approximately 30 trillion body cells in comparison to 39 to 100 trillion microorganisms. The skin covers an outside surface area of approximately two square metres, and the average microbial density is estimated to be 10^6^ per square centimetre [8,9,15]. In reality, the skin covers about 25 m^2^, including invaginations, sweat glands and hair follicles [16]. Actual microbial density differs per body site, with the lowest counts found on the volar forearm, trunk and legs (10^2^/cm^2^) and the highest counts in the underarms, umbilicus and toe web spaces (10^7^/cm^2^) [17].

In general, at least 19 phyla are characterized within the microbiome, of which four phyla dominate the skin: Actinobacteria (51.8%), Firmicutes (24.4%), Proteobacteria (16.5%) and Bacteroidetes (6.3%). Furthermore, three genera represent 60% of the skin microbiome: *Staphylococcus* spp., *Corynebacterium* spp. and *Cutibacterium* spp. [8,13,14,15]. Humans acquire a skin microbiome at birth, but as people grow up, dynamic changes in the microbiome take place [18]. Stabilisation of this microbiome occurs when becoming adults; however, changes as a reaction to environmental factors are not excluded [14,15]. This variation and evolution in the human skin microbiome can be described in four different ways: as topographical, interpersonal, intrapersonal and temporal variation. Topographical variation is defined as the different skin niches, which determine the composition of the present community. As such, a difference can be made between sebaceous (oily), moist and dry niches. In fact, the skin is not an appropriate environment for the bacterial species to live. This is due to the acidic pH, low moisture content, presence of salts and antimicrobial molecules as free fatty acids, sphingosine, nitric oxide, immunoglobulins, etc. Yet, a lot of bacterial species are present on the skin because of this topographical variation. Each topographical niche differs in pH, hair density, sebaceous, apocrine and eccrine activity. This creates microenvironments that select for particular microorganisms. Moreover, some microorganisms have developed specific strategies to survive on the skin [15,16]. Next to topographical variation, there is interpersonal variation, which refers to the differences in skin microbiome between people. Differences in the microbiome within an individual is described by intrapersonal variation. Temporal variation is the change in skin microbiome that might occur within an individual over time.

Healthy skin depends on the diversity of bacterial species that live in close proximity and in relation to one another. The competition between different bacteria constitutes the body’s first line of defence against bacteria and viruses [14]. The microorganisms create an environment that enables the skin to be less susceptible to diseases and pathogenic invasion [19]. A specific example of how bacteria play a role in defence is the production of the Esp protease by *Staphylococcus epidermidis*. This protease inhibits colonisation by the pathogenic *S*. *aureus* [15]. However, besides microbe–microbe interactions, there are also microbe–host interactions. These interactions can be divided into three categories, the first one being commensalism, where one species has an advantage and the other is unaffected. The second type of interaction is mutualism, which can be seen as a win–win situation for both parties. Lastly, a detrimental interaction is an interaction where one species obtains an advantage while adversely affecting the other party. Advantages for the microorganisms include the presence of nutrients on the skin. Therefore, by allowing the microbial community to evolve, the possibility of the skin microbiome to respond in a fast way to environmental changes is improved. As such, the skin forms a better protective barrier, which is, in turn, advantageous for the host [13].

The human skin microbiome faces many challenges. Over the past few decades, there has been an increase in the prevalence of allergic diseases present in Western countries [20]. This links with the ‘hygiene hypothesis’, a theory initiated by Erika Von Mutius, which indicates that children who are kept in very clean environments are more likely to develop hay fever, asthma and a range of other conditions [21]. Environments with high levels of microbial components found in nature can be preventative against the development of allergies long-term [20]. The ideologies on cleanliness, sterility and urbanisation have created a difficult environment for many beneficial organisms to survive on skin [22]. Nonetheless, microbial diversity is the key to a healthy immune system [23]. The skin microbiome has an important role in controlling the skin inflammation and tuning resident T lymphocyte function [19]. The urban environment has made it difficult to re-establish or maintain the beneficial bacteria that the body would regularly be exposed to in a more natural environment [24]. This modern chemistry and shift in lifestyle habits have undermined this ecosystem, making it less diverse and leaving the skin more susceptible to skin pathologies [25].

### 2.3. Odour Formation in Textiles

Clothing textiles are important contributors to odour development. The bacteria in textiles convert sweat secretions, sebum, skin desquamations and other external molecules into volatile compounds. Shelley et al. (1953) stated that odour intensity is potentially more intense in clothes as compared to the axillary skin [26]. Dravnieks et al. (1968) described axillary skin odour as primary odour, and axillary clothing odour as secondary odour [27]. Bacteria attaching and living on and in the fibres use carbon molecules or sebum compounds as substrate and generate volatiles as by-products. This odour can be vastly different from the odour from the axillary skin. There are two main reasons for this: the build-up of sweat secretions in the textiles, and the microbiome converting these molecules into malodour [28].

Three main types of sweat glands are present in the dermis: eccrine, apocrine and apoeccrine sweat glands. The different glands secrete fluids from which composition differs among types of glands [29]. Eccrine sweat glands are the most abundant and can be found on the entire body: on glabrous as well as on non-glabrous surfaces [29,30]. Apocrine glands are the second most important type of glands. The highest concentrations of apocrine sweat glands are found in the underarms and the anogenital area. However, the secretion is viscous and odourless and is composed of proteins, sugars, ammonia, ferric ions, steroids and lipids [8,29,31]. Apoeccrine glands develop during puberty from eccrine-like precursor glands and produce extremely copious sweat, which is secreted directly on the skin surface [29,31,32,33].

Over time, the metabolite load can increase in the fibres, as more sweat and sebum secretions are transferred to the clothes and are insufficiently removed by the laundering process. Specifically hydrophobic molecules, such as apocrine and sebaceous secretions, can adhere and bind to clothes, which leads to a build-up of metabolite precursors [28]. This might be an important explanation for permastink clothes. Not only the load of sweat but also the type of sweat is of influence. It was shown that a mutation in the human ABCC11 gene, which is prominent among east-Asian people, leads to less apocrine sweat secretions [34] and, as such, less malodour build-up in clothes. Additionally, gender, hygiene habits, diet and body mass index can have an impact on odour build-up in clothes.

### 2.4. The Textile Microbiome

The microorganisms living in clothing textiles have been a major target for odour control and disease prevention. The interaction between the wearer and clothing is an opportunity for skin bacteria to attach to the textile surface, which can lead to the growth of certain strains [28]. During this process, the absorption of the wearer’s sweat, sebum and bacterial metabolites to clothing can contribute to the outgrowth of pathogenic strains, contribute to malodour generation or trigger skin diseases [28,35]. The growth of these pathogenic or odour-causing strains are, however, dependent on textile composition, the individual and the activities of the wearer [28].

Bacteria are transferred in high numbers from armpits to clothing fibres due to their close contact. However, axillary and textile microbiomes of one person are not the same and differ in composition. Important odour-causing bacteria in armpits, *Corynebacterium* spp., could not be isolated from worn textiles [36,37]. Particularly low abundant skin microorganisms can selectively enrich in synthetic textiles [38]. Rather, the textile microbiome is composed of *Staphylococcus*, *Micrococcus*, *Bacillus*, *Enterobacteriaceae* and *Acinetobacter*, among others [36]. Selective enrichment of particular odour-associated taxa on different fibre types is an explanation of why some fibres, such as polyester, smell less pleasant. *Micrococcus* has been shown to be enriched on malodorous synthetic clothes, and are known to have the enzymatic potential to cause malodour from apocrine sweat [38,39]. *Moraxella* and *Pseudomonas* were similarly found on washed laundry and associated with malodour formation [40]. Unlike the microbial composition, the microbial load does not seem to be a determining factor in odour formation in clothes [37]. Nonetheless, most techniques to prevent odour or pathogen formation are making use of broad-spectrum antimicrobial agents.

### 2.5. Skin Conditions & Textile Relationship

Our clothing is consistently in contact with the human skin; thus, textiles are an important consideration when studying the cutaneous environment. Therefore, textiles are essential players in the potential causation and treatment of various types of dermatitis and skin diseases [41]. Different factors such as textile structure, pH, breathability, and microbiome should be considered for functional outcomes.

The relationship between textiles and skin is a new focus on how certain textiles and their microbiome can alleviate skin disorders. A good example is for atopic dermatitis, a chronic relapsing skin condition, associated with skin barrier dysfunctions, moisture loss, allergy/immunology, and pruritus symptoms [42]. Atopic dermatitis can be exacerbated by the colonisation of *Staphylococcus aureus* on the affected skin [43]. Atopic dermatitis is in part an environmental-related disease, and textiles and skin contact are an influential part of the cutaneous environment [41]. Research has found that wool and synthetic fibres, such as polyester and nylon, can worsen atopic dermatitis symptoms [41,44,45]. Cotton, however, is a recommended fabric for patients with such skin conditions [46]. Studies have also explored the use of silver-coated, chitosan-coated and cellulose-based textiles for their antibacterial effect on *S. aureus*, and have found promising outcomes [35,47].

The epidermis is characterised by a slightly acidic film, which is called the “acid mantle”, and has a pH value of about 5 or lower [43,48]. The resident skin microbiome is dominated by *S. epidermidis* at a pH of 4.7, and the growth of *S. aureus* is inhibited under these acidic conditions [48]. The skin microbiome is altered if the pH value increases towards a more neutral range. Skin diseases characterised by faulty barriers such as atopic dermatitis, candidal intertrigo and acne are all associated with aberrant pH values [49]. Recent research has explored the possibilities of citric acid-coated textiles as a therapeutic strategy for these chronic skin conditions [43]. In this study, a cellulose-based textile was coated with a citric acid finish in order to lower the pH of the textile surface. It was concluded that the citric acid-coated textiles, when worn next to the skin, reduced symptoms of atopic dermatitis, such as itching, and improved the skin barrier. Therefore, textiles with a lower skin pH might reduce microbial colonisation of atopic skin when worn next to the body. The citric acid-coated fabric might provide a basis for new preventive and therapeutic options in atopic dermatitis and other pH-related skin diseases [43].

Similarly, contact dermatitis is a common skin disorder caused by the direct contact with an agent or surface to the skin. The symptoms are marked by erythematous, vesicular, papular or lichenified pruritic skin lesions [50]. It is caused by irritant triggers in 80% of the cases, such as textiles or chemical agents, and allergic triggers, in the remaining 20% of patients. There are many chemicals used in the production of textiles which have been linked to contact dermatitis. However, disperse azo dyes, frequently used for colouring synthetic textiles and known for its poor adhesion to fabric, have often been found to cause allergic textile dermatitis. This could be due to the lipophilic molecules which can easily migrate onto the skin [51,52].

Other allergens include dust mites, which are found in clothing and bedding, represent a prevalent risk factor for asthma and skin conditions. There are many species of dust mites; however, the most common in homes across the world are *Dermatophagoides pteronyssinus*, *Dermatophagoides farinae* and *Euroglyphus maynei* [53]. There have been many studies linking atopic dermatitis with dust mites. A recent study has linked levels of vitamin D3 and the severity of atopic dermatitis symptoms from dust mite allergies [54]. However, long-term trials of house dust mite reduction methods are needed [53]. In order to remove these allergy-associated dust mites in clothing, it is generally advised to wash fabrics at minimum 55 °C. An alternative for more natural methods for reducing dust mites are being studied. Research on the use of eucalyptus oil on textiles, as a natural alternative to kill the dust mites, is being explored [55].

## 3. Antimicrobial Agents in the Textile Industry

The ability to retain moisture and the large surface area makes textiles contributory to microorganisms’ growth. This microorganism susceptibility can cause a range of undesirable effects on the textile itself and for the user [56,57]. Such effects can include a change in the mechanical strength of textiles, discolouration, a tendency of user contamination and, most commonly, the development of an unpleasant odour [58]. Over the last few years, there has been a developing public awareness of these bacterial effects on textiles and health. More specifically, there is a rising concern about the microbial propagation in textiles used for health care in hospitals. Besides this, with the trend in the current society changing to more active lifestyles, the demand for anti-odour sportswear has been increasing [56]. Therefore, research and development in this field has intensified in order to minimise or even eradicate microbial growth, and a wide range of antimicrobial textiles have been developed [57].

Antimicrobial textiles with odour resistance properties are becoming a new requirement on the market. Odour-resistant textiles can in turn be divided in two other categories: odour-resistant (fragrant) textiles which are used in automotive applications, and odour-resistant technical textiles apparel, which are used in sportswear [59]. The latter contains antimicrobial and/or antifungal compounds. An antimicrobial chemical agent is classified according to the mode of action against the cells’ function. If it initiates the inhibition of cell growth, the microbial agent is biostatic, but if it can kill microorganisms, its effect is called biocidal. However, antimicrobial agents often have both characteristics [56,60,61]. Fabrics provided with such biostatics or biocides can neutralise odours and deodorise the wearer, creating antiseptic environments or even enable textiles to clean themselves [62].

There are two modus operandi that can be applied within anti-odour textiles. The first is the prevention of sweat decomposition by killing the bacteria. For this, several antimicrobial agents have been used: nano silver particles, triclosan, quaternary ammonium compounds, ammonium salts, chlorine derivatives of phenols, drugs, etc. The main synthetic nanoparticle materials and finishes used in the textile industry are discussed in more detail below [59,63]. A second strategy is to chelate chemical, odorous substances originating from the bacterial decomposition of sweat. Examples of chelating molecules are cyclodextrins or calixarenes. An advantage of this technique in comparison to the use of antimicrobials is that it does not affect the skin microbiome. With this technique, however, only odour is combated, and not the source of the problem [59].

There are different possible ways to apply antimicrobial compounds to textiles. First, textile fibres can be impregnated with a solution, suspension or emulsion of the antimicrobial product. Secondly, an insoluble suspension containing the antimicrobial compound can be made. Afterwards, this solution can be applied onto the textile. Thirdly, it is possible to apply the product to the textile through chemical bonding. Finally, during the spinning process, fibres can be immersed in the antimicrobial product during the production process [64].

### 3.1. Synthetic Antibacterial Finishes and Nanoparticles

In the textile industry, biocides are the most used antimicrobial agent, and include products such as metallic nanoparticles and their salts, quaternary ammonium compounds (QACs), triclosan, silver, etc. Nanoparticles in particular are often used in textile finishing because of the ease of incorporation in the fibres [60]. The chemical properties and structure of these components determine the possible modes of action, which are referred to when explaining the different ways of killing the microorganisms. Examples of such modes of action include damage to the cell wall or cell membrane, or either inhibition of the synthesis of these structures, leading to cell leakage and, therefore, cell death. The inhibition of DNA/RNA, protein synthesis or the inhibition of specific metabolic processes within the cell are other possibilities [57].

Different antimicrobials are available to be used as textile finishes, but several requirements are set: (1) it should be non-toxic for humans, (2) it should not give rise to irritation or allergy on the skin, (3) it should be efficient against microorganisms, (4) it needs to be suitable for textile processing, (5) it should be durable during laundering, and (6) it should not impact the quality or appearance of the textile [57].

#### 3.1.1. Metals and Metallic Salts

Metals, such as silver, copper, zinc and cobalt, are often used in the textile industry on fabrics such as cotton, wool, polyester and nylon. Additionally, their equivalent oxides or salts are used as antimicrobial finishes. Metals are known to be extremely toxic in very low concentrations, both in free and bounded states. Another applicability of metals is the use of nanoparticles, which have a higher surface area, resulting in stronger antimicrobials. Nanoparticles are similarly easier to embed into textile fibres [57].

##### Silver

Of all the metals, silver is the best and most used antimicrobial in the textile industry [58,60]. It has broad-spectrum antibacterial properties against Gram-positive and negative bacteria such as *Pseudomonas aeruginosa, Staphylococcus aureus, Staphylococcus epidermidis, Escherichia coli* and *Klebsiella pneumonia.* Even antimicrobial activity against *Staphylococcus aureus* methicillin-resistant strains (MRSA) has been reported [65]. In addition, silver-based antimicrobials are also functional against viruses or even against eukaryotic microorganisms [60]. Silver ions, for instance, bind disulfide or sulfhydryl groups of proteins present in the bacterial cell wall, thereby disturbing cellular processes, leading to death [58]. Other modes of action include interaction with the cell membrane, resulting in the loss of permeability and subsequent intracellular reactions such as DNA condensation, or stimulation of reactive oxygen species (ROS) production, leading to lipid and DNA damage [58]. In the case of silver nanoparticles, microbial toxicity can be controlled by the size, shape and crystalline structure of the nanoparticles. It is known that smaller particles with a size of 1 to 10 nanometers, a triangular shape and a bigger surface area have a higher impact on cells in comparison to bigger particles [60]. An antimicrobial nano silver coating in shirts is a common technique to control odour in textiles. However, it has been observed that these nano silver coatings could affect the skin microbiome in a negative manner (unpublished data).

##### Copper

Copper used as an antimicrobial is comparable to silver as it can also bind disulfide or sulfhydryl groups of proteins in the cell, resulting in cell death [58]. Products such as copper oxide (CuO) are cheaper than silver and have antimicrobial activity against Gram-positive and negative microorganisms. However, higher concentrations of CuO are needed to achieve the same bactericidal effect as silver-based products. Moreover, the synthesis of this antimicrobial is more challenging [60]. Copper ions can also give a characteristic colour to the finished textile. Besides CuO, copper ions or copper salts such as copper sulphate (CuSO_4_) are used in antimicrobial finishes [58].

##### Zinc

Zinc is a trace element that is vital for the immune system, sensory functions and metabolism as well as skin regeneration and protection [66,67]. Moreover, zinc ions, zinc oxides or zinc salts, such as zinc acetate, zinc chloride or zinc sulfate, are also used as antimicrobial finishes [60]. Zinc oxide nanoparticles have antibacterial activity against Gram-positive and negative microorganisms. Bacterial growth is inhibited after the nanoparticle passes through the cellular envelope and disorganises the cell membrane. Moreover, it can generate hydrogen peroxide molecules which damage the cell. These zinc nanoparticles have an equivalent performance as silver or as copper oxide. In a study on zinc nanoparticles and the relationship between activity and size, it was found that the antibacterial activity is inversely proportional to the particles’ size [57,60,68].

As zinc oxide is an essential trace element and has a necessary function in cell renewal and protection of the skin [69], it is said to have soothing and anti-inflammatory capabilities. Since zinc is a component of skin-building enzymes [70], it operates directly on the skin. Zinc has been used to treat numerous dermatological conditions, such as eczema, neurodermitis atopic dermatitis or rosacea [71]; thus, it has been utilised in the textile industry for both its antibacterial properties and skin regeneration.

Additionally, in a study, zinc oxide nanoparticles were applied on cotton and polyester/cotton fabrics and were tested for the protection against UV radiation on the skin. The UV test suggested a significant improvement of UV absorbing activity after the Zn treatment on the textiles. These promising results mean that zinc nanoparticles could be applied to fabric and clothing for the protection of the body against solar radiation, and for their antibacterial properties [72]. Other metal oxide nanoparticles with UV absorption properties include TiO_2_, Al_2_O_3_, CeO_2_ on textile materials [73].

##### Others

There are many different metals that can be used as antimicrobials. Besides the most used metals, silver, copper and zinc—titanium (oxide), magnesium, gold, tin, antimony, zirconium and nickel have also been reported as antimicrobials. However, titanium, tin and semi-metal antimony are not as efficient. Moreover, nickel is less efficient than silver [57,58,60]. The availability and price of these alternative metals can also greatly vary.

#### 3.1.2. Triclosan

Triclosan (2,4,4′-trichloro-2′hydroxydiphenyl ether) is an odourless, synthetic, chlorine-containing derivative of phenol, whereof its antimicrobial activity is concentration- and formulation-dependent. Triclosan is a broad-spectrum antimicrobial effective against Gram-positive and negative organisms and is known to reduce MRSA [57,58,74]. Triclosan has two modes of action: affecting the cell membrane’s integrity by obstructing lipid biosynthesis and/or inhibiting RNA and protein synthesis within the cell. Due to its strong antimicrobial potency, triclosan is often used in the textile industry on polyester, nylon, polypropylene, cellulose acetate or acrylic textiles [57,74]. In recent years, however, it has been associated with harmful side-effects on human skin; therefore, the use of triclosan in textiles is expected to decline [75].

#### 3.1.3. Quaternary Ammonium Compounds (QACs)

In 1930, quaternary ammonium compounds (QACs), a group of 191 components, were discovered to have antimicrobial activity [57,76]. Examples of QACs are hydrophobic, linear alkyl ammonium compounds and their hydrophilic counterparts. The length of the alkyl chain, but also the number of cationic ammonium groups and the presence of perfluorinated groups, determine the degree of antimicrobial activity [57]. Other examples of QACs are cetylpyridinium chloride (CPC), cetyltrimethylammonium bromide (CTAB), benzyldimethylhexadecylammonium chloride (BDHAC), etc. [76,77].

QACs are antimicrobials against Gram-positive and negative bacteria, fungi, including moulds, and some types of viruses [57,58,76,78]. However, Gram-positive bacteria are more sensitive in comparison to Gram-negative bacteria because the latter contains an extra outer cell membrane. Moreover, antibiotic-resistant staphylococci, such as the well-known MRSA, are not sensitive towards QACs as efflux pumps remove the QACs out of the cells [76]. QACs do have different modes of action. The first important target of some QACs is the cell membrane. Quaternary ammonium compounds are cationic and thus are able to bind the negatively charged cell membrane, leading to disorganisation and eventually to cell leakage. In addition, intracellular degradation of nucleic acids and proteins is also possible [57,58,76]. QACs are used as antimicrobials on cotton, polyester, nylon and wool [57].

### 3.2. Alternative Antimicrobials & Therapeutic Fabrics in the Textiles Industry

#### 3.2.1. Environmental and Toxicity Concerns

The antimicrobials used today for textile finishing bring up several concerns. Firstly, antibiotic resistance is a global problem that humanity faces today [57,58]. Gaining resistance is an evolutionary process that occurs within microbial cells. Microorganisms acquire genes leading to antibiotic resistance mechanisms such as the development of efflux pumps, for instance, to avoid antimicrobials entering the cell. As such, MRSA is not sensitive to quaternary ammonium compounds [76]. Moreover, resistance is a trait that can be interchanged between cells. Besides resistance against QACs, resistance against silver particles has also been reported. This means that antimicrobials are losing their efficiency in killing microorganisms, and therefore, in the context of antimicrobial finishing, losing their ability in reducing odour [57].

Antimicrobials can have negative health effects as well [79]. Biocidal materials can have a detrimental effect on the skin microbiome and potentially worsening body malodour in the long term [80]. These synthetic finishes tend to provide a short-term solution to prevent malodour in textiles and do not tackle the root of the problem [62].

There are also concerns about the permeability, neurotoxic effects, and potential toxicity, especially in the early development of the used antimicrobials [75]. In this context, especially triclosan is of concern as it is known to be an endocrine disruptor and is able to move through human tissues. Trace-elements of triclosan can be detected in organisms, and while it is known that triclosan causes oxidative stress, apoptosis, and inflammation in cells, it remains to be determined what the continuous, long-term and low concentration exposure on (human) cells can cause. Regarding metallic compounds, it is already known that there is a risk of neurotoxicity [57].

Next to health effects or resistance problems, the environmental impact of antimicrobials should also be considered. Sustainability is an important topic driving us to think about ecological and environmentally friendly alternatives if products bring up implications for the environment. In the case of silver, triclosan and quaternary ammonium compounds, there is a danger of wash-off during laundering of the textiles [57,58,60]. Triclosan is even discouraged to be used in textiles or clothing that should be laundered regularly. The usage of antimicrobials is especially a problem when they cannot be removed during wastewater treatment. As such, antimicrobials can be toxic and harmful to aquatic organisms [79]. Removal of antimicrobials can be achieved via sedimentation or biodegradation. Silver and triclosan can be removed efficiently; however, if triclosan is degraded under anaerobic conditions, more toxic byproducts are formed. Contrarily, quaternary ammonium compounds are not metabolized or degraded very well. However, due to the positive charge, absorption to negatively charged particles such as sludge, soil or sediments is possible [57]. Nonetheless, not only the antimicrobials themselves but also the textile treatment process has an impact on the environment. It is clear that it is time to shift towards “green technologies” and “green antimicrobial agents” which are both renewable and sustainable [57].

#### 3.2.2. New Directions

Some proposals to reduce the impact on the environment have been made. An example of this is the application of finishes that are delayed in the release of active ingredients, which would also increase the performance of antimicrobial textiles. Besides this, the usage of binders or cross-linking to apply the antimicrobials onto textiles makes the antimicrobials more resistant to the laundering process.

To tackle the problem of antibiotic resistance, in particular, odour management control should rather focus on selective inhibition of odour-associated bacteria instead of having broad-spectrum effects. Recent investigations have shown that microbial load is not necessarily the problem in odour formation in textiles; but rather the microbial composition [28,37]. There is, however, more research needed to identify the odour-associated or pathogenic bacteria and find novel ways to selectively inhibit them.

Another possibility for both the resistance and environmental issues is the use of “green antimicrobial agents”, for instance, natural antimicrobials such as chitosan, alginate, starch or antimicrobials extracted from plants or herbs [60]. Examples of the latter are terpenoids, flavonoids, and tannins. These natural antimicrobials are environmentally friendly and have several advantages such as safety, easy availability and nontoxicity to the skin, which is described in more detail below. Moreover, no antimicrobial resistance is known for these natural compounds [57]. However, the search for new antimicrobials is continuing and other promising molecules, such as antimicrobial peptides (AMPs), are further investigated. AMPs are small peptides with an amphipathic structure, built of 12 to 50 amino acids, that have a broad-spectrum microbial activity. AMPs affect RNA and DNA synthesis, but also induce membrane damage or even loss of ATP molecules within cells [57,76].

### 3.3. Natural Antibacterial & Therapeutic Finishes

The widespread use of antibiotic-coated surfaces has been linked to the emergence of multi-drug-resistant bacteria. Additionally, the polluting effects of the textile industry starts from the manufacturing of the fibre itself. The cost of synthetic antimicrobial finishes to the environment is becoming more obvious, and at the same time, natural dyes and finishes in the textile are gaining significant momentum [81]. This line of interest is motivated by the strict environmental standards imposed by many countries due to the overuse of chemicals that can cause toxicity to humans and nature. This direction of research has emerged in the field of textiles and apparel technology through utilising natural materials, on account of their compatibility with deodorising properties [81]. The research into natural or naturally derived finishes is an important step toward toxic-free manufacturing. The ecology of production methods, waste disposal and the effects of the materials on the human skin are to all be considered [82]. This increased focus on natural antibacterial finishes in the textile industry has led to some new innovations and reestablishing the old. Here we describe the main natural antibacterial fibres and nanotechnologies used for odour control and skin disease prevention.

#### 3.3.1. Bamboo

In China, bamboo fibre is known as “Air Vitamin” or “long-lived element”. It contains anions that are helpful in purifying blood, calming the nervous system, relieving allergy symptoms, and according to Munjal and Kashyap (2015), this is beneficial to the health of the human body and skin [83]. Bamboo is a natural product, containing cellulosic fibre, hemicellulose, ash and lignin [84]. It is one of the fastest-growing plants, which grows effectively without pesticides, fertilizers or herbicides [82]. The main manufacturing methods in the production of sustainable bamboo textiles are chemically based and mechanically based processes [85]. Yet, the fibre is completely biodegradable and is a solution for many of the environmental problems in the textile industry, as its manufacture from raw materials to decomposition does not cause harmful effects to the environment [82]. It has been suggested that bamboo fibre has a unique antibacterial bio-agent known as “Bamboo Kun”. However, “Kun” directly translates to the hydroxyl functional group (–OH), yet the exact chemical compound and origin of this antibacterial property was not clear until more recently [86,87]. As with cellulose and hemicellulose, bamboo is a complex material consisting of hydrophobic lignin and hydrophilic carbohydrates, which are chemically bound [86,88]. It was concluded in a 2011 study on the origin of the antibacterial function of bamboo that the antibacterial compound of bamboo is located in lignin [86]. Lignin contains aromatic and phenolic functional groups which are thought to be responsible for bamboo’s antimicrobial activity.

An additional function of bamboo fibre includes its high absorbency properties which wick sweat away from the human body very efficiently. The absorbent properties of bamboo are said to be due to the microholes in the structure of bamboo caused by the cross-section of bamboo fibres, which is being utilised in the clothing and textiles industry [89]. As such, bamboo clothing is growing in the textile market with a claim for its antimicrobial properties and eco-friendly manufacturing systems.

#### 3.3.2. Chitosan

Chitosan ((C_6_H_11_O_4_N)_n_) is a linear polysaccharide of β-1,4-glycosidic bounded D-glucosamine and *N*-acetyl-D-glucosamine units. It originates from deacetylated chitin, which in turn can be extracted from the cell walls of several fungi or from crustaceans such as crabs and shrimps. Chitosan has an antimicrobial activity because of the cationic properties of its primary amino groups under an acidic pH. As such, chitosan can interact with the negatively charged microbial cell membrane of a range of microorganisms [57,90]. Moreover, the structural characteristics of chitosan, such as its molecular weight, degree of polymerization and deacetylation, actively affect its antimicrobial activity [90]. Environmental conditions, such as temperature, pH or ionic strength, also have an impact on chitosan’s antimicrobial properties [91]. There are methods of improving the antimicrobial properties of chitosan, such as including a permanent positive charge through chemical modification of the chain structure. Other antimicrobial organic compounds, including essential oils and metallic nanoparticles, have also been combined with chitosan [90]. These adjustments to the material propose to advance chitosan’s solubility and antimicrobial activity without altering its biodegradability and biosafety properties [92].

Chitosan’s film-forming abilities and mechanical strength make it an ideal material for food packaging and biomedical devices. However, due to its abundance and desirable properties such as biodegradability, there has been a focus on the use of chitosan in textile for its unique antimicrobial bio-based properties [90]. Nowadays, several chitosan-based products are already commercially available. It is mainly applied as a functional additive in cotton, polyester and wool [57].

The fashion industry is beginning to make use of chitosan’s naturally antibacterial properties in textiles. Often, the natural biopolymer is extracted from crustacean shells that are a by-product of the fishing industry [93]. The anti-odour properties, derived from its presence in crustacea, help inhibit bacteria on the textile surface. The use of chitosan’s bio-based antimicrobial properties is intended to reduce the need for washing, thereby helping to lower the garment’s overall carbon footprint.

#### 3.3.3. Seaweed Fibre

Medicinal plants and bioactive organisms have been used as a natural remedy for human diseases for centuries; yet, more recently, there has been an increased interest in these materials for their pharmacological and low-toxicity properties on the body [94]. There has been a decided shift in chemical-based ingredients to biodegradable and sustainable products with health and wellness properties [95].

Seaweeds or algae are autotrophic organisms, which consist of varied species and are found in many different marine habitats globally [96]. Marine algae can be classified into three different classes which can be distinguished by their difference in pigment, cell wall composition, and reserve polysaccharides; Phaeophyceae (brown), Rhodophyceae (red), and Chlorophyceae (green) [92,97]. These differences in seaweed classes hold varied properties, yet they all acquire a phenolic compound in its functional group, such as flavonoids, which contribute to the wide biological activities, including antioxidant, antiviral and antimicrobial activities [98,99].

The bioactive nature of seaweed has emerged as an important natural ingredient in recent years. In order to make use of this trend, research has focused on isolating and identifying the bioactive compounds and constituents from marine algae to apply in various industries [96]. The use of active ingredients in textiles for therapeutic or cosmetic purposes is an emerging field [94]. Seaweed is rich in vitamins, minerals, essential amino acids and proteins, which have known therapeutic benefits to the skin and body [100]. Such properties can improve the immune system of the skin, encourage cell renewal, and hold antioxidant, anti-inflammatory and antimicrobial capacity [101]. These benefits could, in turn, help to relieve symptoms of skin conditions such as neurodermatitis, atopic dermatitis or psoriasis and inhibit the formation of free reactive oxygen species, which can alter cell membranes and attack the genetic material of cells [102,103].

Investigations into a viscose and algae composite fibre have been developed for the textile industry to make use of these skin care properties. Materials from sea algae have been utilised for their soft skin feel, and natural abundance in vitamins, micronutrients and elements. The fibre has been tested for its antioxidant capacity when the textiles are worn next to the skin [104].

#### 3.3.4. Vitamin-Encapsulated Fibre and Nanoparticles

The encapsulation of vitamins into textile fibres has been developed in recent years, namely for the compound’s important role in skin health. The diverse chemical structures of these compounds are essential for the normal growth and development of the human body. The main function of vitamins is their ability to work as biological catalysts and structural materials for important enzymes, which enables various biochemical reactions for the skin and body [105].

Vitamin E has been a focus in the cosmetic and skincare industry for its bioactive properties and association with skin health. The group of lipophilic compounds that make up vitamin E comprises four tocopherols and four tocotrienols. Vitamin E has a chemical structure which is constructed of a two-ring 6-hydroxychroman structure with an assembly of three isoprenoid units [106]. The lipid-soluble antioxidant properties of vitamin E are known to contribute to the protection of the epidermis and dermis against oxidative stress influenced by the environment [107]. Due to its antioxidant properties (its ability to inhibit the formation of free radicals), it can protect against lipid peroxidation and has the ability to slow skin ageing [108]. Vitamin E contributes to the healing of wounds, and the treatment of dermatological conditions such as subcorneal pustular dermatosis, atopic dermatitis, psoriasis, acne vulgaris, scleroderma; and in the prevention of skin cancer [109]. In a 2002 study, the relationship between vitamin E intake, IgE levels and symptoms of atopic dermatitis were explored. The results suggested vitamin E to be a promising therapeutic treatment for the disease [110]. The relationship between vitamin E and skin health is being explored in the textile industry; vitamin E loaded fabrics have been tested for their antioxidant effects on the skin through microencapsulation processes or the integration into the matrix of the yarn [111]. The vitamins were applied to textiles using a controlled release microencapsulation system. It was concluded that it protects the skin and has anti-aging and moisturising properties. However, further in vivo studies are recommended to verify its effects on skin [111].

#### 3.3.5. Essential Oils Nanoparticles

Due to the large surface area and high surface energy of nanoparticles, they are very efficient for usage in nano finishing of textiles. These benefits can harbor improved durability of overall textile function and greater affinity to the fabrics [89,112]. As such, the nanoencapsulation or finishing of essential oils into textiles has attracted more attention in recent years as a biobased alternative. Essential oils have anti-odour or antimicrobial properties and do not produce major side effects, as is the case with synthetic active agents. Essential oils, which are mainly composed of terpenes, are made up of a complex mixture of volatile compounds and are often found in aromatic plants [113]. In the natural world, these ‘essential’ oils play a role in the protection of the plants because of the antibacterial, antiviral, antifungal and insecticides properties [89,112]. Since ancient times, plant extracts from peppermint, lemon, clove and cinnamon have been used because of their antibacterial, inflammatory and antioxidant activity on the skin. Due to these properties, these plant extracts can be used in wound healing products [114,115]. Other examples of natural oils used for their antibacterial properties are neem oil, castor oil, karanja oil and citronella oil. Advantages would be the efficacy and absence of any adverse effects; consequently, there is an increased focus on applications that make use of these oils [116]. Additionally, pomegranate, orange, and lemon peel are presently the subjects of study for the possible environmentally friendly textiles [117].

An investigation into coated fabric shows antibacterial properties against Gram-negative and Gram-positive bacteria, indicating that this technique can be used in the textile industry as an antimicrobial finish of medical clothes, sports and leisurewear to inhibit odour in apparel [118]. Peppermint oil has been incorporated into textiles to make use of its anti-odour and low toxicity properties. The textile industry utilises the oils anti-odour and antimicrobial properties of peppermint, by finishing the textile using padding technology [119]. The incentive is to provide an alternative to toxic antimicrobials and to reduce the need to wash clothing due to the anti-odour activity of the finished textile.

#### 3.3.6. Hydrogen Peroxide

Hydrogen peroxide (H_2_O_2_) is capable of inhibiting invasive bacteria on textiles. As this compound is produced by human and bacterial cells, hydrogen peroxide is also classified as a natural antimicrobial. The antimicrobial activities of hydrogen peroxide can be used to inhibit bacteria, viruses, fungi, including moulds, and it can also be found in honey, acting as a preservative [79]. Additionally, certain concentrations of H_2_O_2_ are often used to remove staining on textiles, odour and improve textile deterioration [120]. Such finishes can accomplish antimicrobial effects in various textile products, which include apparel, upholstery and automotive industries [79,121]. Its main drawback is the instability of the molecule, meaning there is a need to stabilise the molecule or use it as fast as possible.

## 4. Microencapsulation and Emerging Technologies

Traditionally known for their ability to absorb moisture, innovation in textile fibre technology reveals that they can release substances too. Transdermal drug delivery is a method used in the healthcare sector in which medication can be administered directly through the skin via a textile’s surface [122]. This kind of technology has enabled fibre and textile manufacturers to develop new concepts for health, wellness, and odour control. Beneficial molecules and even microbes can be microencapsulated and attached to the textiles for sustained release. Vitamins, probiotics, antimicrobial ingredients, enzymes, fragrances, and others are common examples of products that utilise this technology.

### 4.1. Incorporating Beneficial Bacteria in Textiles

Bacterial and fungal species are known for the ability to decompose organic matter (sweat, blood, saliva, organic waste, amongst others) with the use of specific enzymes. Species such as *Bacillus, Lactobacillus, Enterobacter, Streptococcus, Nitrosomonas, Nitrobacter, Pseudomonas, Alcaligens* and *Klebsiella* spp. are already used in odour control. A specific application of this is for carpets, for instance. After spilling organic matter on a carpet, dormant microorganisms are activated and degradation of the spill by enzymatic actions will prevent odour development. Other products in which bacteria are used to counteract odour are pillows, mattresses, towels, footwear, automotive products, body protective gear, etc. [123]. Bacterial usage in textiles for anti-odour is a growing field of research.

Novel approaches are desired as a result of the increased demand for multipurpose and sustainable textile materials. Research into probiotic bacteria as an alternative for typical biocides and antimicrobials are being developed to reduce the current toxicity and resistance problems. For example, the antimicrobial effect of kombucha has been studied for its effects against pathogenic bacteria and candida strains. The antimicrobial activity observed in the fermented broth was significant against Gram-positive and negative pathogenic bacteria, including *S. aureus* and *Candida* strains [124]. These are promising findings that have the potential to provide an alternative to synthetic antimicrobial drugs. The Food and Agriculture Organisation (FAO) and World Health Organization (WHO) have defined probiotics as “living microorganisms that provide a positive health effect for the host” [125,126]. However, if probiotics are used, it is important to be sure that the bacteria stay alive on the substrate. The viability of the bacteria affects the efficiency of the product. To protect and guarantee bacterial viability, microencapsulation of the cells is often carried out in industrial applications. This principle is often used in the food industry for the production of probiotics and the use in dairy products [125].

Over recent years, the use of microcapsules, or the microencapsulation of active ingredients for personal care and textiles products, has developed considerably. This process means active substances can be fixed either to the surface of the fibre or within the fibre itself depending on its composition. The aim of microencapsulation is to develop a coating protecting the core active ingredient from the environment. In fact, there are two groups of microparticles: those who are built of a core and surrounded by layers forming a protective shell or microparticles where the active ingredient is embedded in a matrix [127].

Microencapsulation technology processes are advancing in the textile industry with a focus on the incorporation of active ingredients to fabric surfaces. These new fabric concepts include textile products incorporating microbiota in fibres which mainly focus on their anti-allergic benefits. The process encapsulates probiotic spores into a textile to significantly decrease the dust mite allergen. This technology is predominately used for incorporation into mattresses where the spores are released with friction. The encapsulation process uses a polymer matrix to form shells around the active ingredients [128].

Studies have incorporated probiotics into textiles in order to reduce infectious pathogens that can be acquired in hospitals. The institution experimented with an adapted sol–gel coating process to embed these probiotic beneficial spores on a woven polyester surface. The viability of bacteria was analysed along with the physical properties and characteristics of the coated fabric. The results demonstrate a successful encapsulation of the beneficial spores with a sufficient number of living organisms, before and after repeated washing cycles, as well as suitable tensile strength and abrasion resistance properties. The surface wettability remains an area of improvement in order to maintain adequate adhesion between substrate and coating [129]. Other investigations have explored the use of spore-forming *Bacillus* spp. in cleaning agents to reduce odour on (textile) surfaces [130]. These bacilli have also been tested for odour management in footwear for athletics and leisure. This process of applying live microorganisms to fibres results in natural, non-toxic health benefits to consumers and the environment [131]. There are many strains of probiotic bacteria that can be associated with different benefits inside the body and on its surface. Thus, the incorporation of probiotics into textiles has not been fully assessed, yet they provide a promising technique to combat pathogenic bacteria on textile surfaces in the future [132].

### 4.2. Emerging Technologies

Skin microbiome research has become a hot field in skincare in the last few years, namely through means of bacteriotherapy for malodour control. This research is also being acknowledged by the textile industry and new technologies are being developed continuously. Synthetic biology has provided some clues as to the future direction of performance wear and the second skin. Biotechnological and engineering methods will help to expand the functions, properties and potential of textiles surfaces along major frontiers [12]. Thereby, functions in order to treat diseases and injuries will be developed. Along with the biotechnological advances, other future technologies include bioinspired nanostructures, graphene and microbiome-smart clothing, which are described below.

#### 4.2.1. Bio-Inspired Nanostructures

In recent years, there has been a new approach for realising bactericidal action through physical surface topography. Certain natural nanostructured surfaces have been identified to have the capacity to rupture the cell wall of the bacteria, which is known as the “contact killing mechanism” [133]. This research is inspired by certain insects which are inherently known for their bactericidal surfaces are able to kill microbes on contact. This bactericidal effect is due to the sharp nanostructures, being 80 to 250 nm in height and having a 50 to 250 nm nano-pillar and a 100 to 250 nm pitch, present on the insect’s exterior surface [134]. These sharp structures have the potential to pierce the cell wall of the bacteria, therefore rupturing its surface and killing the bacteria. This physical biocidal approach has become a new area of focus for tackling multi-antibiotic resistant bacteria [134,135].

Various techniques have been used to fabricate nanostructured surfaces on a variety of substrates including silicone and polymer surfaces, which have been tested against different pathogenic bacteria [136,137]. However, the scope of the bactericidal activity of such nanostructured surfaces depends on several criteria including the shape of the structure, the size and density. While finding an optimised nanotopography technique, material and process remain a challenge, such developments have presented opportunities for further investigations [134,138]. Yet, results have shown it to be a promising strategy in inhibiting pathogenic bacteria to solve antimicrobial resistance challenges in the future [134].

#### 4.2.2. Graphene

Graphene’s key components are carbon atoms organised in one-atom-thick planar sheets packed in a crystal lattice. Other graphene materials include graphene oxide and reduced graphene oxide. Besides its self-cleaning ability, flame retardancy, electrical conductivity and UV-blocking characteristics, graphene is known for its antimicrobial properties [139]. Graphene’s antimicrobial features can be controlled by different parameters: its lateral size, number of carbon layers, shape, surface modifications and agglomeration ability. Moreover, different modes of action are already suggested. First of all, graphene can act as a so-called nano knife because of its sharp edges, which breaks up the cell membrane, giving rise to leakage of intracellular compounds. Secondly, graphene is able to induce oxidative stress, disturbing essential functions within the cell, causing cellular inactivation or even cell death. Thirdly, because of the thin structure of graphene, it is said to wrap around a bacterial cell blocking nutrient transport towards the cell. Besides these three main mechanisms, other active modes such as extraction of membrane lipids or interference in protein–protein interactions are described as well [140]. Yet, research regarding graphene and its antimicrobial properties is still at its infancy.

In a recent study, a graphene/TiO_2_ nanocomposite was applied on a polyester fabric in order to obtain electroconductive, antistatic, hydrophobic and UV protective properties. The graphene provided electrical conductivity and the presence of titanium resulted in good protection against UV. The UV properties are an important consideration since prolonged exposure to ultraviolet rays are linked to skin cancer [141].

#### 4.2.3. Microbiome-Smart Clothing

New research is exploring the impact of the textile microbiome and points out the direct involvement of particular microbes on malodour development. Research is investigating so-called “microbiome-smart” textiles, in which good microbes, or their enzymatic potential are added, or the microbiome is steered towards non-odour-causing and healthy communities.

The microbiome plays a key role in body odour and antibacterial fabric finishes on clothing have been designed in an attempt to reduce odour issues. In the past decades, innovation has mainly focused on broad-spectrum antimicrobial technologies. Yet, these are associated with many drawbacks. Research is now looking into the incorporation of healthy skin commensal bacteria into textiles (Broadhead et al., unpublished). The microorganisms are activated in contact with the moisture on the skin, allowing them to dominate other less beneficial bacteria. The probiotic clothing technology provides an alternative to antibacterial chemicals in clothing and cosmetics and a solution for natural odour management.

The COVID-19 pandemic and the worldwide necessity of mask wearing has been linked to changes in the skin microbiome and exacerbated skin diseases. The frequent use of face masks has led to a significant increase in acne flare-ups on the skin (so-called “maskne”). The increase of this inflammatory condition could be due to an increase in humidity and friction while wearing the mask. The sebum accumulation on the skin from the sebaceous glands can aggravate an existing condition or prompt outbreaks [142,143].

Antiviral surfaces have become an innovation focus since the pandemic, particularly in the textile industry. Studies have looked at the possible transmission routes of SARS-CoV-2 and one hypothesis is that mites that have been associated with human skin infestation, such as *Demodex* or *Pyemotes* species, could play a role in the spread of the virus. The lipid bilayer that encloses the SARS-CoV-2 virus has the ability to remain stable on sebum-rich skin and therefore provides another route of infection. The exoskeletons of these arthropods common on skin share a common polymer, chitin or chitosan. The material discussed earlier in this paper is known for its non-toxic, biodegradable, biocompatible properties to the skin. These unique properties have meant a fresh focus on chitin as a scaffold to build nanoparticles for enhanced transdermal drug delivery application [144,145].

The study suggests arthropod–coronavirus interactions could develop through the molecular forces between the chitin found on mites and the lipids present on the exterior of SARS-CoV-2. This may mean that mites on the skin are an important factor in viral infection, which may result in important biomedical or textile developments for both prevention and treatment [144,145]. As a result, research is going into alternatives for the current antibacterial, antiviral and antiparasitic finishes. Such collaborative effort in science, technology and fashion aims to use the recent advances in science into a novel approach in the textile industry.

## 5. Conclusions

Laundering habits changed in the 1990s towards more ecologically friendly conditions (lower temperature, less water use and absence of chemical detergents). This is a promising development but has also led to reduced microbial and odour removal. An increase in synthetic fibres in textile production has exasperated odour retention in textiles and pathogenic colonisation of bacteria, which can trigger skin diseases. As a result, we saw the rise of a wide array of antimicrobial finishes in clothes, which are not always effective and are sometimes associated with unwanted side-effects. Recently, particular odour-controlling agents have been encapsulated in nanoparticles to show a sustained effect.

The relationship between body odour and skin diseases, textile and the skin microbiome is an emerging field of research. The skin microbiome is shaped by its natural environment and what is put on and next to the skin has an immediate impact. Designing microbiome-smart textiles can be a novel and alternative way to advance the functionality of clothing and to combat odour development or potential textile-related skin conditions. To make a shift in the effects of the antibacterial ingredients and toxic cosmetics that society has encouraged, the answers may be to look more closely at the skin’s living ecosystem and natural skin biome. The knowledge of the microbiome and their responsibility in odour formation in textiles is slowly but steadily growing. Research has shifted towards more natural and bio-inspired applications to control odour formation in textiles. As research digs deeper into the fundamental origins of malodour formation, pathogenic overgrowth in textiles are developed to combat these bacterial-led side effects in clothes, enhance the skin feeling and advance the overall well-being of the wearer.

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
