# Peer review of "The Future of Functional Clothing for an Improved Skin and Textile Microbiome Relationship"

_microorganisms, 2021, doi:10.3390/microorganisms9061192_

Round 1

Reviewer 1 Report

1.Line 70,mites as Demodex Folliculorum and Demodex Brevis should be add.Also the endosymbionts of Demodex like B Oleronius,Simplex,Cereus,Pumilus,Kronstadtii,etc should be mentioned,referenced and their influence on the facial microbioma added.

2.Chapter 2.3 explanaitions about eccrine and apocrine sweat glands are wellcome to be added also the apocrine activity after puberty should be described and the odour implications before puberty discussed

3.Information about the contact dermatitis to antiseptics,antimicrobials from textiles should be add ,comment and referenced

4.Add information and references about other possible future natural antimicrobials for textiles like Komboucha,pasture

5.After 4.2.2 part some information about UVB non penetrating textiles and their possible usage for skin protection combined with textile antimicrobials should be added.

6.Before Conclusions,informations about the mask wearing ,facial microbioma,scalp textiles for the cold season are of interest for the readers.

7.Information about facial chitin lipid interactions between Demodex and viral particles,the possible action of some textile to be prepared for scabies or pediculosis,so with a antiparasitary action is of future interest and should be discussed and referenced

Author Response

Dear Reviewer,   After our correspondence by email, we are submitting a revised version of our review entitled “The Future of Functional Clothing for an Improved Skin and Textile Microbiome Relationship”. We would like to submit this review to the Special Issue of Dr Lionel Breton on “Microbiome Interorgans Axis (MIA): A Future Option in Health and Diseases”. This review describes the skin and textile interphase and their microbiome impact, as well as an exhaustive list of techniques and novel ways to improve functional clothing.   Microbiome-smart textiles will be the new frontier in functional clothing and is here explained for the very first time. The study focuses on dermatology, skin microbiome, functional clothing and textile innovation. The review is relevant for any studies related to those fields.   After the reviewer's comments, we have discussed the relationship with clothing and mites and future research on its contribution to spreading Covid-19. We described the different sweat glands on the skin and the apocrine activity after puberty. We have expanded section 2.5, where we also discuss Contact Dermatitis and potential triggers in textiles. Kombucha is discussed as a future method in textiles for its antimicrobial properties. UV protection is an important aspect in maintaining skin health, therefore UV protective Zinc and Titanium nanoparticles studies have been referenced in the manuscript. Regarding future research in textile and skin, we have added studies that discuss the relationship between mask-wearing and skin conditions.    The reviewer comments were of valuable input and have made the paper better! Thank you for assisting in this process.
  Please see attached the Rebuttal letter.    Best regards, Chris Callewaert

Reviewer 2 Report

  1. 111 (Scharschmidt and Fischbach 2013)

need to remove italics

  1. 331-332 “Zinc nanoparticles have a size of about 30-40 nanometers which is inversely proportional to the particles’ antibacterial activity”

This is an incorrect citation of the original source. Zille et al. compare the sizes of titanium oxide and zinc oxide nanoparticles and the concentrations at which nanoparticles are effective.

  1. 332-334 “There are a lot of different metals which can be used as antimicrobials. Besides the most used ones being ………….antimoni”

Antimoni is not metal It Is a semi-metal

  1. 371 “QACs are antimicrobials against Gram-positive and negative bacteria, fungi, moulds”

Moulds are fungi, so it's better to write: fungi, including moulds

  1. 539-540 “Marinealgae can be classified into three different species”.

Phaeophyceae, Rhodophyceae and Chlorophyceae are Classes

  1. 543 “These differences in seaweed species hold varied properties”

Classes.

  1. 627 “inhibit bacteria, mold, fungi, viruses”

Should be: “fungi, including moulds”

Author Response

Dear Reviewer,

After our correspondence by email, we are submitting a revised version of our review entitled “The Future of Functional Clothing for an Improved Skin and Textile Microbiome Relationship”. We would like to submit this review to the Special Issue of Dr Lionel Breton on “Microbiome Interorgans Axis (MIA): A Future Option in Health and Diseases”.

This review describes the skin and textile interphase and their microbiome impact, as well as an exhaustive list of techniques and novel ways to improve functional clothing. Microbiome-smart textiles will be the new frontier in functional clothing and are here explained for the very first time. The study focuses on dermatology, skin microbiome, functional clothing and textile innovation. The review is relevant for any studies related to those fields.

After the reviewer's comments, we have discussed the relationship with clothing and mites and future research on its contribution to spreading Covid-19. We described the different sweat glands on the skin and the apocrine activity after puberty. We have expanded section 2.5, where we also discuss Contact Dermatitis and potential triggers in textiles. Kombucha is discussed as a future method in textiles for its antimicrobial properties. UV protection is an important aspect in maintaining skin health, therefore UV protective Zinc and Titanium nanoparticles studies have been referenced in the manuscript. Regarding future research in textile and skin, we have added studies that discuss the relationship between mask-wearing and skin conditions. The reviewer comments were of valuable input and have made the paper better! Thank you for assisting in this process.

Please see attached the Rebuttal letter.

Best regards,

Chris Callewaert
